# A Nanotechnology-Based Approach to Biosensor Application in Current Diabetes Management Practices

**DOI:** 10.3390/nano13050867

**Published:** 2023-02-26

**Authors:** Ambreen Shoaib, Ali Darraj, Mohammad Ehtisham Khan, Lubna Azmi, Abdulaziz Alalwan, Osamah Alamri, Mohammad Tabish, Anwar Ulla Khan

**Affiliations:** 1Department of Clinical Pharmacy, College of Pharmacy, Jazan University, Jazan 45142, Saudi Arabia; 2Department of Medicine, College of Medicine, Shaqra University, Shaqra 11961, Saudi Arabia; 3Department of Chemical Engineering Technology, College of Applied Industrial Technology, Jazan University, Jazan 45142, Saudi Arabia; 4Department of Pharmaceutical Chemistry, Institute of Pharmaceutical Sciences, University of Lucknow, Lucknow 226025, India; 5University Family Medicine Center, Department of Family and Community Medicine, College of Medicine, King Saud University Medical City, Riyadh 2925, Saudi Arabia; 6Consultant of Family Medicine, Ministry of Health, Second Health Cluster, Riyadh 2925, Saudi Arabia; 7Department of Pharmacology, College of Medicine, Shaqra University, Shaqra 11961, Saudi Arabia; 8Department of Electrical Engineering Technology, College of Applied Industrial Technology, Jazan University, Jazan 45142, Saudi Arabia

**Keywords:** biosensors, glucose biosensor, nanomaterials, nanotechnology, diabetes management

## Abstract

Diabetes mellitus is linked to both short-term and long-term health problems. Therefore, its detection at a very basic stage is of utmost importance. Research institutes and medical organizations are increasingly using cost-effective biosensors to monitor human biological processes and provide precise health diagnoses. Biosensors aid in accurate diabetes diagnosis and monitoring for efficient treatment and management. Recent attention to nanotechnology in the fast-evolving area of biosensing has facilitated the advancement of new sensors and sensing processes and improved the performance and sensitivity of current biosensors. Nanotechnology biosensors detect disease and track therapy response. Clinically efficient biosensors are user-friendly, efficient, cheap, and scalable in nanomaterial-based production processes and thus can transform diabetes outcomes. This article is more focused on biosensors and their substantial medical applications. The highlights of the article consist of the different types of biosensing units, the role of biosensors in diabetes, the evolution of glucose sensors, and printed biosensors and biosensing systems. Later on, we were engrossed in the glucose sensors based on biofluids, employing minimally invasive, invasive, and noninvasive technologies to find out the impact of nanotechnology on the biosensors to produce a novel device as a nano-biosensor. In this approach, this article documents major advances in nanotechnology-based biosensors for medical applications, as well as the hurdles they must overcome in clinical practice.

## 1. Introduction

Diabetes mellitus (DM) is considered one of the most commonly occurring chronic ailments. It has been a constant cause of morbidity and mortality and continues to grow at an alarming rate [1,2]. It is divided into several kinds, with type 1 and 2 DM being the most prevalent. Type 1 DM is characterized by an inability to produce insulin due to T-cell-mediated autoimmunity that destroys pancreatic cells [3]. In contrast, type 2 diabetes is characterized by insulin resistance and a decline in insulin levels. The most widespread kind of diabetes is type 2, which affects adults and occurs when the body becomes resistant to insulin or does not generate enough [4,5]. Diabetes mellitus afflicted 422 million people globally in 2014, up from 108 million in 1980. In 2012, excessive blood glucose was responsible for a further 2.2 million fatalities [5,6]. There was a 5% rise in diabetes-related premature death between 2000 and 2016. It is projected to have caused 1.5 million deaths in 2019, and it is estimated that this figure will drastically rise to 552 million by 2030 [7]. By 2045, an estimated 629 million individuals worldwide will have DM, with the highest predictions in developing nations [8,9]. Regarding the incidence of DM, Saudi Arabia is ranked second among gulf countries and seventh in the world, with an estimated 3.4 million of its population suffering from it. The latest estimate has shown that nearly 24.4% of adults in Saudi Arabia suffer from DM [10]. Between 1990 and 2015, the incidence of type 2 diabetes mellitus (T2DM) in Saudi Arabia ranged from 18.5% to 31.6% depending on the region [11]. Ever since the discovery of oil, the Kingdom of Saudi Arabia has experienced extraordinary economic growth and development; this has been evident in the last few decades. The westernization of the culture has brought about drastic changes in the diet and led to a lot of changes in eating practices [12].

The primary markers in diagnosing and managing this condition are glucose and insulin. Controlling blood glucose levels can help postpone and, in some cases, avert problems. As a result, blood glucose monitoring is essential in managing diabetes [13]. Type 2 diabetes treatment often emphasizes lifestyle management. Thus, these patients must be focused on exercise and weight control, with a medical diet, oral glucose-lowering medications, and insulin injections being typical treatments [14]. However, the main target is to find out the best technique for the determination of blood glucose levels.

In the past few years, there has been a surge in demand for inexpensive, disposable, user-friendly, cost-effective, and mass-produced diagnostic medical devices with quick response times. A glucometer is a standard tool for determining blood glucose levels. This approach is becoming less successful, as well as highly costly. Analytical instruments with a biological sensing aspect are known as biosensors. These biosensors have gained popularity as a means of continuously monitoring blood glucose levels to maintain them within normal limits [15,16]. They demonstrate sensitivity and specificity with complicated analytical measures, making them easier to run. Glucose biosensors represent about 85 percent of the worldwide biosensor market. However, transmembrane, nucleic acid, and whole-cell or cell network affinity sensors have garnered attention [17]. Signal transduction-based high-throughput screening is becoming more important. Biosensors have evolved into the most sophisticated instrument for detecting glucose and insulin, and they come in various forms [18]. This article covers electrochemical, optical, enzymatic, non-enzymatic, noninvasive, and real-time biosensors. In recent years, there has been progress in creating nano biosensors by utilizing different nanomaterials, increasing their application in various sectors. We have looked at how to make insulin and glucose biosensors for diabetic neuropathy treatment, modify them, and establish new techniques (Figure 1).

## 2. Different Parameters for the Diagnosis of Diabetes

Utilizing implanted bioelectronics to evaluate physiological indicators has lately become a medical treatment or diagnostic tool [19]. Diabetes, diabetic foot, heart disease, renal failure, and stroke are all conditions that can lead to death. Early-stage diabetes can occur without clinical signs, making symptomatic patients difficult to identify. For diabetes prevention and early treatment, continuous urine monitoring and blood glucose monitoring are required in daily life [20].

### 2.1. Urine Glucose Monitoring

Elevated glucose levels in urine are considered dangerous since they suggest the progression of diabetes. A positive urine result shows glucose levels in the system are greater than 50–100 mg/dL (2.78–5.55 mM) [21]. In 1996, the first Japanese urine glucose meter significantly monitored urine glucose levels from 0 to 500 mg/dL [22,23]. In 1999, the TOTO Corporation released a urine glucose monitor device that was incorporated into toilet seats. The built-in meter was used to monitor diluted urine samples (Figure 2).

Carbon nanotubes (CNTs) have also been investigated to detect glucose in urine. Dissolved CNTs in biopolymer chitosan (CS) aqueous solutions allow urine glucose measurement without interference (with a detection limit of 3 M) [3,24]. Kimura Group’s UG-201 is an extremely sensitive urine glucose meter that uses an Amperometric glucose sensor. It has outstanding performance characteristics, such as a broad measuring range of 0–2000 mg/dL, a rapid reaction time of 6 s, and resistance to interference from interferents, such as ascorbic acid/acetaminophen. There was an excellent linear association between the developed urine glucose meter and a commercially available urine glucose analyzer for clinical application. The developed urine glucose meter was used to monitor post-meal blood glucose levels by assessing the urine glucose of actual people. The new portable meter was created to use locations other than the house or office [4].

According to research published in 2016, ZnFe_2_O_4_ magnetic nanoparticles (MNPs) with in-built peroxidase-like activity could be used as a colorimetric biosensor for detecting glucose in urine. For glucose detection utilizing glucose oxidase (GOx), ZnFe_2_O4 MNPs are affordable, highly sensitive, and selective, with a linear range of 1.25 × 10^−6^ to 1.875 × 10^−5^ mol/L (1.25–18 µM) and a detection limit of 3.0 × 10^−7^ mol/L (0.3 µM) [25]. Using an electronic nose, Siyang et al. suggested a new approach for detecting diabetes based on a direct assessment of urine odor (e-nose). The e-sensing nose’s parts were made up of eight commercial chemical gas sensors. Cluster analysis (CA) and principal components analysis (PCA) approaches were used to study the data. The method successfully determined glucose levels in urine [6].

Wang et al. created a smartphone ambient-light sensor and a label-free colorimetric test for determining urine glucose (ALS). Quantitative H_2_O_2_ was applied to samples to determine the deepest color using the horseradish peroxidase hydrogen peroxide 3,3′,5,5′-tetramethylbenzidine (HRP-H_2_O_2_-TMB) system, a smartphone ambient-light sensor was attached to assess transmitted light illuminance, and color variations were used to compute urine glucose concentration [26].

Zhang et al. developed a flexible self-powered biosensor device connected to a diaper to detect urine composition. The device was powered by an enzyme biofuel cell (EBFC), which generated energy using glucose from urine as fuel. The biosensor system, which consisted of EBFC, a power management system, and a light-emitting diode, was a self-powered sensor that could detect glucose levels in diabetes patients’ urine without the use of external power [27]. Lee et al. developed a paper sensor composed of polyaniline nanoparticles (PAni-NPs) [28] and noninvasive, intuitive, and highly selective red blood cell membranes (RBCMs) [29]. The glucose transporter-1 protein in RBCM (coated on PAni-NP-adsorbed paper) functions as an intelligent filter, transferring glucose while rejecting other interfering molecules. The RBCM-coated PAni-NP-based paper sensor was about 85 percent more selective than uncoated paper sensors. The paper sensor could detect 0.54 mM and could detect urine glucose concentrations between 0 mg/mL and 10 mg/mL (0–56 mM). The creation of a highly specific colorimetric urine glucose monitoring device is made possible by this paper sensor [30].

### 2.2. Blood Glucose Monitoring

Due to the enormous number of disease-related markers and the less intrusive collection process, blood has been the most extensively utilized sample type. As a result, it becomes a useful supply of biomarkers for biosensors that may be carried around [31]. Monitoring fasting plasma glucose levels is inadequate for achieving effective post-meal glucose management [32]. Recent research suggests that lowering post-meal plasma glucose is critical for achieving reduced hemoglobin A1c (HbA1c), a marker of diabetes progression [32]. The self-monitoring of blood glucose is extensively proven as an efficient strategy for controlling diabetic blood glucose levels, and much work has gone into designing blood glucose sensors [33].

#### 2.2.1. Glucometer

As previously indicated, commercial glucose sensors rely on puncture tests that should be performed up to seven times daily. Abbott Diabetes Care Ltd. Omron Healthcare, Inc. and Roche Diagnostics Ltd. are among the companies that sell finger-pricking sensors for personal usage (Accu-Chek). Glucometers, a well-known and advanced technology that gives accurate results quickly and with high sensitivity, are the most commonly used conventional sensors for managing glucose [34]. The blood glucometer was initially presented in 1970 as a semiquantitative and visual instrument for estimating glucose levels by matching a pad’s glucose-specific response to a printed color scheme [35]. Even though the initial blood glucometer needed numerous stages, a considerable blood volume, and precise timing, they could help people with diabetes regulate their blood glucose levels. The redox potentials of the two enzymes (GOx at −48 mV vs. SHE at pH 7.2 and GDH at 10.5 mV vs. SHE at pH 7.0), stability, turnover rates, and glucose affinity and selectivity are all different [36]. GOx has a more robust selectivity for glucose than GDH and can survive more significant pH, ionic strength, and temperature fluctuations. GOx, on the other hand, catalyzes glucose oxidation at a rate of 5000 s^−1^, compared with GDH’s rate of 11,800 [37]. After glucose is oxidized, a mediator, such as ferrocene derivatives, hexacyanoferrate, or quinones, carries the enzyme signal to the working electrode.

Despite the commercial success of contemporary mediator-based BGM systems, it has been shown that some medications, metabolites, and other blood components may interact with these glucose sensors [38]. Blood glucometer devices available today are adequate for monitoring glucose levels in diabetics; however, 95% of the measured glucose readings must be within 15 mg/dL (0.83 mM) of the reference measurement at glucose concentrations of 100 mg/dL (5.6 mM) or within 15% at glucose concentrations of 100 mg/dL (5.6 mM), according to the ISO standard [39]. Given the variety of variables that might affect glucose measurements, such as hematocrit, temperature, altitude, and human error [40], one of the few commonly used in vitro diagnostic devices that have been upgraded with cellular capabilities and mHealth apps is the blood glucose meter.

#### 2.2.2. Colorimetric Strip

Xue et al. developed a glucometer that is flexible, self-powered, and skin-like to monitor blood glucose levels in the body in real-time for the prevention and treatment of diabetes. The operative mechanism is based on the coupling effect between piezoelectricity and enzyme reactions in arrays of GOx@ZnO nanowires. Under strain, the device converts mechanical energy into piezoelectric impulses. Blood glucose levels affect this process. The piezoelectric voltage generates electricity and biosensing signals. The implanted device can check blood glucose levels live [19].

As a non-glucose quantitative portable detection system, Li et al. proposed a unique design that combines a classic blood glucose monitoring device using a lateral flow strip and a commercially available smartphone (iBGStar^®^ Blood Glucose Monitoring System). An oxidative DNA impairment biomarker, 8-hydroxy-2′-deoxyguanosine, is used to demonstrate the notion (8-OHdG). The device’s fundamental design is based on a gold nanoparticles (AuNPs)-based competitive immunoassay, as described in the colorimetric visual detection platform. Visual detection, on the contrary, provides only qualitative and somewhat quantitative information. Quantitative analysis is made possible by switching from target detection to enzyme invertase detection with this method [41].

#### 2.2.3. Ames Reflectance Meter

In 1970, Anton H. Clemens created the first blood glucose meter, the Ames Reflectance Meter (ARM), which used reflected light from a Dextrostix strip to determine a person’s blood sugar level [42]. A significant drop of blood (about 50–100 L) was put into the reagent pad, which was gently rinsed away after one minute, and the pad color was visually analyzed against a color chart to provide a semiquantitative blood glucose measurement. Reflocheck, a tiny portable reflectance meter that used Reflotest strips that were cleaned with a cotton ball and included a barcode for calibration, was introduced by Boehringer Mannheim (BM) in 1982. The findings of a diabetic screening trial in general practice revealed cost savings, and the evaluations showed remarkable correlation and precision [43].

### 2.3. Glucose Biosensors

#### 2.3.1. The Clark Enzyme Electrode

The most popular interstitial fluid (ISF) analysis technique for clinical continuous glucose monitoring remains implanted enzyme-electrode sensors [44]. Because the active working electrode (WE) is commonly made on the cylindrical-end face, this type of implantable sensor requires a tiny diameter such that the WE area is very small. The sensor’s sensitivity is limited due to the narrow WE area. As a result, typical enzyme–electrode sensors struggle to detect hypoglycemia, which is a severe disease [45]. To reduce signal drift and enable hypoglycemia diagnosis, a new cylindrical, flexible enzyme–electrode sensor with a larger WE surface is presented for implanted continuous glucose monitoring. A cylindrical substrate was used because it retains the most surface area for a given volume compared with other geometries, and a larger cylindrical surface creates a larger WE surface. By passing the diameter restriction imposed by conventional pin-like enzyme–electrode sensors on a cylindrical surface, the WE may be formed not only along the radius but also along the axis of the cylindrical substrate [45].

Attaching the appropriate oxidase enzyme to the tip of a Clark-type oxygen microelectrode resulted in microsensors for glucose. The enzyme is immobilized in a polyacrylamide matrix on the electrode tip before being covered with a polyurethane membrane. The quantity of oxygen used by the electrode, and hence the biosensor’s output, is controlled by the analyte concentration in the sample. These micro biosensors had tip sizes ranging from 15 to 40 m, reaction durations ranging from 0.5 to 5 s, and could detect as little as 2 m of the analyte. These microsensors proved a versatile instrument for monitoring particular analytes in unstirred settings, with a spatial resolution of 100 µm or less and speedy response times [46].

#### 2.3.2. Yellow Springs Instrument

The Yellow Springs Instrument (YSI) 2300 STAT PLUS Glucose and l-Lactate Analyzer (YSI 2300) is the most extensively used comparator instrument for manufacturers to calibrate batches of glucose test strips during manufacturing for self-management blood glucose devices. This gadget has also been frequently used in investigations to show that continuous and exploratory noninvasive glucose monitors are accurate [47].

#### 2.3.3. Mediated Biosensors

The ultimate objective of glucose monitoring is noninvasive glucose sensing, and the following are the fundamental techniques being investigated for glucose sensor development [48]. Despite the relatively simple use, rapidity, and low danger of infection associated with infrared spectroscopy, this technology is vulnerable to frequent calibrations, poor selectivity, limited sensitivity, and miniaturization challenges. The lack of a correlation between ejected fluids and blood glucose concentrations is one of the concerns, with direct glucose monitoring using void physiological fluids. Changes in glucose concentrations in the fluids caused by exercise and nutrition can also lead to erroneous readings [49]. The ambition to construct an artificial pancreas motivates researchers to keep working on biosensors. However, the shortcomings of in vivo biosensors before creating such an insulin-modulating gadget must be addressed [50].

A variety of glucose sensors are available, most of which are intrusive to the patient. It has been shown that fiber optic sensors provide advantages over conventional sensors and have great application potential, notably in the healthcare sector [51]. Compared to other sensors, they are smaller, more manageable, and mainly noninvasive, resulting in a reduced risk of infection, as well as being accurate, well-correlated, and inexpensive. Different devices based on other approaches, such as optical polarimetry [52] and electrode detection [53], and fluorescence with devices created for glucose management beneath the skin, such as the Senseonics Eversense, have been developed alongside this technology.

In fluorescent glucose testing, the intensity or duration of signal decay may be assessed. The fluorescence lifetime is different for each analyte, which may be evaluated in dispersion media [54] and which aids in distinguishing between substances. Water absorbs less radiation in spectroscopy; hence, light may permeate through the epidermis stratum corneum to generate a higher blood concentration regardless of skin color, making it the material of choice for building noninvasive glucose sensors [55]. Electromagnetism-based technology, on the other hand, monitors glucose using voltage or current based on magnetic coupling [56]. This process depends on the current or voltage connection between the output and input, which is directly proportional to the glucose concentration.

## 3. Biosensors for Diabetes—A Special Case

The World Health Organization (WHO) presently identifies diabetes as one of the primary chronic disorders endangering human health [57]. People with diabetes have unpredictable blood glucose levels and are at risk for developing serious ailments, including diabetic neuropathy, stroke, blindness, renal failure, and cardiovascular difficulties [58]. Patients with diabetes must routinely check their blood glucose levels and often inject insulin if their glucose levels are too much or ingest sweet items if their glucose levels are less. Unquestionably, a wearable device that continuously monitors blood glucose would greatly simplify and improve blood glucose control, lowering the likelihood of complications.

For the treatment and control of diabetes, it is thus essential to create a system for precise glucose detection. A biosensing and -detecting converter and peripheral electronic equipment can monitor diabetes patients’ glucose levels in real time. The sensing system relies on biosensing detection conversion, which impacts several essential parameters, including detection time, accuracy, and system cost [59,60,61]. In comparison with conventional detection technologies, a biosensor offers the benefits of rapid analysis, high accuracy, cheap cost, excellent repeatability, straightforward operation, and high specificity [62,63,64,65]. Chips, microfluidics systems, and labs on a chip have helped biosensor technology advance rapidly. Depending on the signal conversion, biosensors can be optical or electrochemical [66,67,68]. By putting particular enzymes on the electrodes, electrochemical biosensors may detect biomarkers. They have various benefits, including high sensitivity, ease of use, and cost effectiveness [67,69,70]. Electrochemical biosensors react with glucose in the blood to produce gluconic acid. By monitoring the current signal, gluconic acid concentration and, therefore, the blood glucose level, may be calculated. Blood glucose levels that are considered normal for a person who does not have diabetes should be between 4 and 5.9 mM/L (before meals) and under 7.8 mM/L at all other times (2 h after a meal) [71].

Nevertheless, the insertion of some small-molecule exotic vectors that function as conduits among enzymes and motor electrodes results in a lousy performance, slow response, frequent replacement, and reduced dependability. In addition, another important obstacle restricting the adoption of electrochemical sensors is that the electrolytes must be frequently refilled, which substantially increases the burden of eventual expenditures [72,73]. Optical biosensors need a long settling period for detection and are very vulnerable to ambient-light-altering test findings [74,75]. From history, it is clear that several biosensors are introduced and implemented from time to time (Table 1) [36,75].

There has been a lot of buzz about a biosensor that uses radio frequency (RF) as of late, and it is being held up as a potential and powerful competitor to third-generation glucose biosensors [18,88,89]. RF biosensors provide the following benefits compared with other biosensor types: Firstly, they vary from electrochemical biosensors, which are susceptible to the limitations of the usage environment and performance deterioration with time. The RF biosensors are not subject to environmental elements, such as temperature, external light, and humidity, and they can retain stability in long-term, complicated situations. Second, the RF biosensor does not need to be pre-stabilized and can detect biomarkers in real time. Finally, the RF biosensor’s detection time is independent of anything but the vector network analyzer’s sweep period, demonstrating a clear benefit in detection time (Table 2 and Table 3; Figure 3).

### 3.1. Printed Biosensors and Biosensing Systems

When glucose binds to the enzyme glucose oxidase, a change in the signal strength at a nearby electrode is proportional to the concentration of glucose in the sample. Traditional “hard” electronics are notoriously difficult and expensive to produce because of the electrode materials themselves (such as, lithography). Sensors whose performance criteria are secondary to cost effectiveness, including disposable and one-time-use sensors, are typically fabricated using additive printing processes (e.g., screen printing and inkjet) and materials compatible with large-area deposition. Inkjet printing is a low-temperature method that permits the controlled deposition of various electronic materials in customizable shapes [117]. When it comes to integrating biological molecules, such as enzymes, with the electronics needed for biorecognition [118,119], it is necessary to make use of this property [118,119]. Recent developments in organic electronic materials (e.g., conducting polymers) facilitate printing processes. Due to its excellent conductivity and compatibility with screen printing [120] and inkjet printing [121], poly (3, 4-ethylene dioxythiophene): polystyrene sulfate (PEDOT: PSS) is the standard conducting polymer for printed electronics [122]. PSS-based devices have been employed as biosensors for obtaining cutaneous recordings of electrophysiological activity [123,124], as well as for the enzymatic sensing of gases [121] and glucose [119]. Because of its gentle character, PEDOT: PSS may also be printed on recyclable, inexpensive, and eco-friendly paper substrates that replace costly glass and plastics, paving the way for disposable in vitro diagnostic instruments [125].

### 3.2. Implantable Biosensors and Noninvasive Monitoring

A biosensor implanted for long-range, continuous glucose monitoring is particularly important for reducing the risk of acquiring diabetes and its complications [126]. Several investigations have revealed implanted sensors for continuous glycemic monitoring to alleviate the discomfort caused by enzyme-based finger pricking [127]. A 32 mm volume biosensor and an enzymatic electrochemical method detected glucose concentration [128]. In addition to an electrochemical biosensor, a transponder chip must also be powered. While extremely useful, these sensors are still unreliable; at the heart of the problem lies the difficulty of implanting an active device in the body. Even though implanted glucose monitoring devices provide frequent glucose level measurements, this technique is not advised for all people with diabetes owing to its intrusive nature [19], and specific continuous glucose monitoring methods have been observed to be up to 21% inaccurate [129]. Initially, these sensors required replacement every three days and provided a measurement every five minutes after a two-hour run-in period. In 2006, another American business, Dexcom, introduced a sensor with a seven-day lifespan [130]. These mistakes are often ascribed to sensor drift caused by enzyme catalytic efficiency variations. This necessitates a frequent recalibration of the instrument through the finger-prick procedure [131]. On the other hand, recent improvements in implanted sensors may be utilized to include insulin pumps, allowing for immediate insulin delivery [127].

Sode et al. have also created the Bio-Radio-Transmitter, a self-powered implanted device for continuous monitoring that may be used in an artificial pancreas. In this case, the gadget consists of a capacitor, radio receiver, and radio transmitter [132]. The capacitor of the Bio-Radio- in the presence of glucose causes the transmitter to emit a radio signal, which is picked up and amplified by the receiver. The variation in frequency is then linked to blood glucose levels [132].

The ultimate objective of glucose monitoring is noninvasive glucose sensing using near-infrared spectroscopy, enzyme electrodes, microcalorimetry, optical sensors, and iontophoresis and sonophoresis, which extract glucose from the skin and are the principal glucose sensor development methods [48]. Notwithstanding its relative simplicity of use, quickness, and low danger of infection, infrared spectroscopy is hampered by poor selectivity, limited sensitivity, frequent calibration requirements, and issues with downsizing. There is little association between voided fluids and blood glucose concentrations, which is one of the problems with direct glucose measurement using physiological fluids. Exercise and food that affect glucose concentrations in fluids can result in erroneous findings [49]. The goal of constructing an artificial pancreas is the driving force behind ongoing biosensor research. However, the disadvantages of in vivo biosensors must be eliminated before developing such an insulin-modulating device. Several noninvasive ways to measure blood glucose have been described [133,134].

## 4. The Impact of Nanotechnology in the Development of Biosensors for the Detection of Diabetes

Historically, glucose biosensors have been used in the medical field to manage diabetes. Glucose levels in the blood, which remain a key predictor of diabetes, are monitored using biosensors [78]. Biosensors have remained vital because they enable humans to manage blood sugar levels and evaluate disease-related ecological responses. In addition to providing healthcare and treatment, biosensors can expeditiously diagnose diseases and monitor patients’ situations. For the previous two decades, biosensors were needed for environmental monitoring, clinical diagnostics, and identifying biohazardous bacteria and viruses [135,136]. In vitro biosensors based on organic organisms have complicated nanofabrication-enabled biosensors. Thus, the recent emphasis on nanotechnology in the fast-evolving area of biosensing has facilitated the development of novel sensors and sensing mechanisms, improving the performance and sensitivity of current biosensors [137,138,139,140]. The employment of biosensors in the everyday diagnosis of diseases such as diabetes exemplifies the widespread diagnostic applications of these technologies [141].

Recent breakthroughs in nanotechnology have allowed us to construct structures and devices in the nano-domain, such as nanoscale particles, nanorods, nanotubes, and nanowires, which directly interact with the biomolecular targets we aim to detect using biosensors [140,141,142,143]. These devices demonstrate exceptional electrical conductivity and improved optical, electrical, and magnetic characteristics compared with traditional biosensors, and they have the potential for increased sensitivity and quicker reaction time [141]. Nanotechnology has helped with these efforts by making sensors with more surface area, improving the catalytic properties of electrodes, and making sensors on the nanoscale [144]. Bioengineering’s recent technical advances offer an array of applications, ranging from biosensors to medication delivery.

With the application of nanotechnology, including electromechanical, resonant, thermal, magnetic, and optical approaches, there have been significant advancements in illness detection [141,145]. The development of nano-biosensors has significantly improved the capacity to detect particular analytes and gain detailed information on the biomolecular outlines of various illnesses. Due to the accurate sensing and analysis of specific biomarkers, improved biosensor systems that detect changes in localized micro-environmental elements have been created [145]. This provides a sign of the illness, its course, and the effectiveness of treatments. Despite the fascinating capabilities of these nano-biosensors, they suffer from some drawbacks, including drift, fouling, non-specificity, and the presentation of non-uniform and irreproducible signals. Measuring glucose levels in the blood is still one of the best ways to predict who will develop diabetes. Therefore, in the medical field, glucose biosensors have been used to help predict diabetes [10]. Nanoscale materials, such as GNPs, CNTs, magnetic nanoparticles, Pt nanoparticles, quantum dots, and others, affect glucose sensor performance. The fibrous shape and wrapping of PDDA over MWCNTs load GOx into the electrospun matrix. Electrodeposited Pt nanoparticles on the MWNTs matrix were simple and reliable. Chitosan-SiO_2_ gel can immobilize glucose oxidase on Pt/MWNTs electrode surfaces. High-sensitivity serum glucose biosensors were created [146].

## 5. Nanomaterials as Potential Additives to Existing Biosensors

A nano-biosensor uses nanotechnology to measure, monitor, and analyze biological events [147]. These devices usually contain quantum dots, nanoparticles, nanowires, or nanofilms. Most biological systems, such as membranes, viruses, and protein complexes, interact in the nanoscale range [148,149]. Therefore, nanoscale devices are good candidates for biomedical and bioanalytical applications due to their high sensitivity, specificity, and quicker reaction times than conventional biosensing techniques [150]. There are a variety of applications for nanostructured materials, such as the use of Amperometric nanodevices for enzymatic glucose determination, quantum dots for binding determination, and even bioconjugated nanomaterials for specific biomolecular detection. For example, colloidal nanoparticles have been conjugated with antibodies for precise immuno-sensing purposes [148,151]. A similar approach has been used to detect and analyze DNA/RNA by utilizing metal nanoparticles’ electronic and optical properties [152]. The critical components of nano-biosensor systems include nanoparticles, quantum dots, and nanotubes. This structure has enabled devices, such as nanoprobes, nano-sensors, and other miniature systems, to revolutionize chemical sensing and biology [152].

In such nanodevices, reaction times and power consumption are high, while the device size is small [153]. The electrochemical signals created at the electrode/electrolyte interface have been enhanced by employing nanomaterials, such as oxide nanoparticles, metal nanoparticles, magnetic carbon materials, nanomaterials, metal phthalocyanines, and quantum dots [139,154]. The use of functionalized nanoparticles bonded to organic molecules for biosensors has been developed. Several different methods can make these nanostructured materials and nanodevices depending on the nanomaterial or material of interest (either 0D, 1D, or 2D), their size and quantity, and the synthesis method [154].

Recent research has focused on developing diagnostics and biosensors using nanomaterials, such as graphene, quantum dots (QDs), carbon nanotubes (CNTs), nanocomposites (NCs), and metal nanoparticles (M-NPs) (Table 4). As seen in Figure 4, exploring nano-biosensors might include a variety of research topics.

## 6. Hurdles for Biosensors in Clinical Practice

First and foremost, biosensors must be very selective if they are to be used in medical diagnosis. The validity of a test depends on how well it isolates the signal caused by the analyte of interest from background noise. An enzyme or antibody is a common biological specifier used in diagnostic tests. The enzyme glucose oxidase has long been established as this specifier’s target for glucose; more recently, glucose dehydrogenase has also been utilized with effectiveness [172].

### 6.1. Hurdles for Composition and Secretion of Epidermal Biofluids Such as Simultaneous Monitoring of Sweat (ISF) Monitoring

The devices presented in this article are especially beneficial for fitness monitoring since they satisfy the criterion of sweat generation during physical effort. Epidermal devices need innovative sample pathways to work better (for example, diabetes monitoring or alcohol monitoring). Wearable biosensing devices need to perform the noninvasive monitoring of novel target biomarkers to improve range and efficacy [173].

### 6.2. Hurdles in Wearable Tear-Based Biosensors

Biosensors based on tears have been widely used for glucose monitoring, but they also offer enormous promise for the noninvasive detection of other physiologically important markers. Other metabolites and critical electrolytes whose concentrations in tears are closely connected to those in the blood may be added to the list of new tear analytes, for example, direct, noninvasive tear-based catecholamine assays may assist in the diagnosis of glaucoma) [174].

### 6.3. Specialty Hurdles in Biosensors

Medicine has long used biosensors, including commercial devices such as the glucometer, which revolutionized blood glucose monitoring and enhanced the quality of life for millions of diabetics, helping the field the most [175]. However, P4 medicine—personalized, predictive, preventative, and participatory—demand is increasing.

Despite the very promising scenario of using biosensors as analyzers in strategic science and technology, limitations remain and represent serious problems in this area. The following should be considered in the field of biosensors for advanced use:
Optimizing biosensors’ sensitivity, specificity, and accuracy for rapid serological tests (such as for infectious diseases, etc.);Improving the production efficiency of disposable, inexpensive, and highly efficient biosensor devices;Improving the platforms and performance of wearable and investable biosensors.


## 7. Conclusions

Due to the increased incidence of diabetes mellitus and the need to avoid its consequences, there has been a significant rise in the use of glucose monitoring devices and the need for revised recommendations for the care of diabetic patients. Detection and diagnosis are essential to successfully treat patients for various human diseases at the onset of their progression. To detect disease, biosensors must be simple, sensitive, and affordable. The medical field has wide applications for biosensors, which can help doctors and patients, such as for illness control, clinical care, treatment, patient health monitoring, and disease research. Many applications for nanomaterials have been found through the development of biosensors in recent years. Biosensors are easy to use in clinical medicine, and their purpose is to stratify patients according to their condition based on those indicators. It is widely recognized that biosensors enable personalized medicine, a new medical practice approach in the modern era. This approach to healthcare enables the availability of several treatment and diagnostic options. The advantages of low invasiveness and continuous glucose monitoring in new medical devices might increase patient comfort and understanding of their glycemic condition, improving the healthcare system and overall clinical results.

## 8. Future Perspective

Continuous glucose monitoring that does not require invasive procedures might be an alternative that holds a lot of appeal and promise for the way the field is headed in the future. This approach to healthcare enables the availability of several treatment and diagnostic options that can be used as an effective novel technology for the treatment and management of diabetes and related disorders. Overall, a glucose biosensor with high sensitivity, good stability and selectivity, low cost, a long shelf life, and suitability for industrial manufacture is anticipated to be developed shortly as a result of the ongoing development of research methodologies and improvements in technologies and processing.

## Figures and Tables

**Figure 1 nanomaterials-13-00867-f001:**
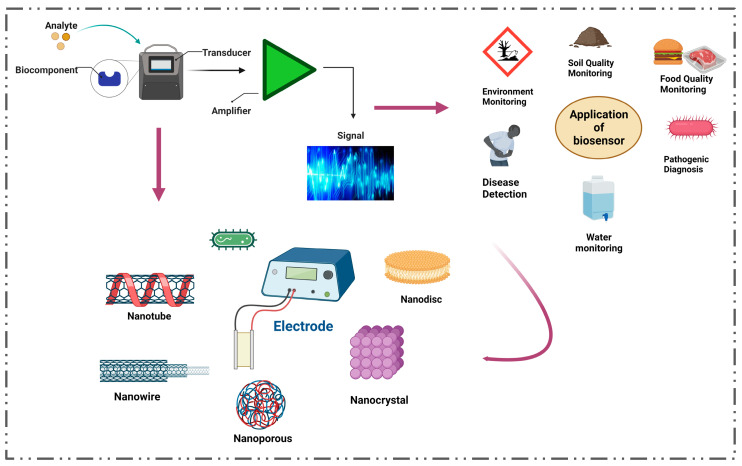
Biosensor for advanced biotechnology and research applications.

**Figure 2 nanomaterials-13-00867-f002:**
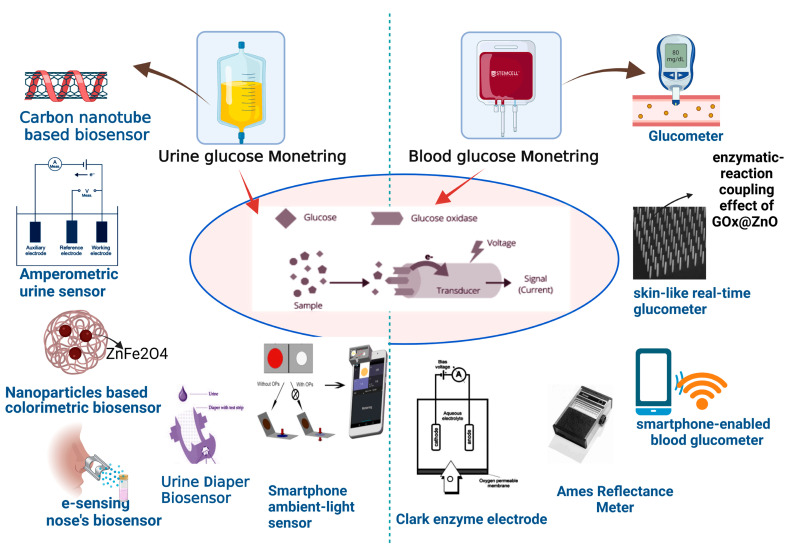
Diagnosis of glucose level in urine and blood with help of modern biosensor tools.

**Figure 3 nanomaterials-13-00867-f003:**
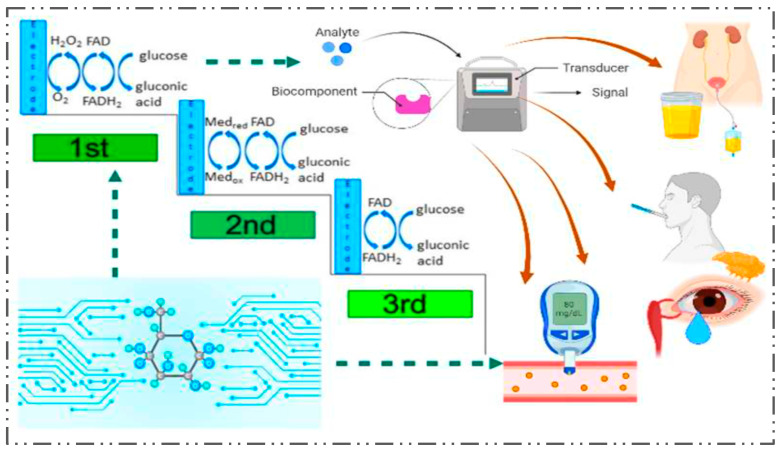
Illustration of glucose biosensor from the first to the third generation and its action on various biological fluids, such as urine, saliva, tear, and blood.

**Figure 4 nanomaterials-13-00867-f004:**
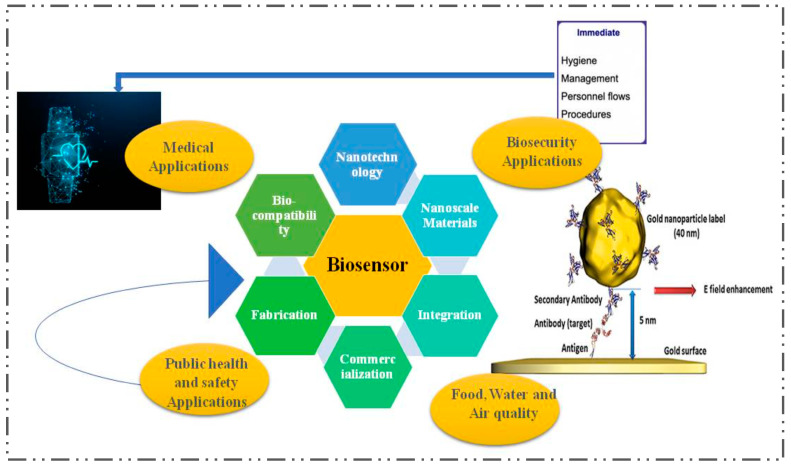
Nano-biosensors: Research focused on nanotechnology.

**Table 1 nanomaterials-13-00867-t001:** The evolution of glucose sensors.

Year	Events	References
1962	The Children’s Hospital of Cincinnati’s Clark and Lyons provided the first description of a biosensor.	[76,77]
1967	Updike and Hicks developed the first functional enzyme electrode.	[36]
1973	Electrode for measuring glucose based on hydrogen peroxide detection.	[78]
1975	The first commercial biosensor reintroduced after a hiatus, i.e., Yellow Springs Instrument Company analyzer (Model 23A YSI analyzer).	[36,79]
1976	The world’s first fully functional artificial pancreas for use in the hospital (Miles).	[36,80]
1982	Shichiri made the first needle-shaped enzyme electrode for implantation under the skin.	[81]
1984	Cass made the first glucose biosensor based on ferrocene and amperometry.	[36,82]
1987	The MediSense ExacTech blood glucose biosensor is now on the market.	[36]
1999	Commercialization of an in vivo glucose sensor (MiniMed).	[83,84]
2000	The first noninvasive glucose meter that can be worn (GlucoWatch).	[36]
2010	Subcutaneous self-powered sensor, Paper-based sensor.	[36]
2015	Non-enzymatic sensors; Extensive use of noninvasive biofluids; on-body patch sensor.	[36]
2022	Enzyme-free glucose sensors utilizing two metal nanoparticles and a ferrocene derivative for effective synergistic electrocatalysis; a flexible enzyme-free glucose sensor fabricated from leaf-inspired micro/nanostructures; silk-based electrochemical sensor.	[83,84,85]
2023	Continuous glucose and lactate monitoring with NIR Luminescent Oxygen Nanoparticles; wearable glucose biosensing with platinum and nickel hydrogels; carbon nanotube for electrochemical glucose sensors; glucose biosensors with chemo-optical sensing microdomains; wearable noninvasive glucose sensor.	[85,86,87]

**Table 2 nanomaterials-13-00867-t002:** Glucose sensors based on biofluids employing minimally invasive technologies.

S. No.	Assay Method	Minimal Sample Volume (µL)	TEST Time (second)	Assay Range (mg/dL)	Hematocrit Range (%)	Memory	Manufacturer	Brand	References
1	GOD	0.3	5	20–600	25–60	400	Nova Biomedical	Nova Max	[90]
		1.0	5	20–600	30–55	500	LifeScan	OneTouch UltraLink	[91]
		0.5	4	20–600	20–60	300	AgaMatrix	WaveSense KeyNote	[92]
		1.4	8	20–600	30–55	300	Bionime	Rightest GM300	[93]
		0.7	7	20–600	20–60	450	Diabetes Supply of Suncoast	Advocate Redi-Code	[94]
		0.6	6	20–600	20–60	450	Diagnostic Devices	Prodigy Autocode	[95]
2	GDH-PQQ	0.6	5	10–600	20–70	500	Roche	Accu-Chek Aviva	[96]
		0.3	5	20–500	15–65	400	Abbott	FreeStyle Freedom Lite	[97]
3	GDH-FAD	0.6	5	10–600	0–70	480	Bayer	Ascensia Contour	[98]
4	GDH	0.3	5	10–600	30–52	360	Arkray	Glucocard X-meter	[99]

**Table 3 nanomaterials-13-00867-t003:** Glucose sensors relying on biofluids and making use of noninvasive technologies.

S. No.	Assay Method	Sample	Limit of Detection Range	Limit of Detection	Noninvasive Biosensor	References
1	Electrochemistry	Saliva	2 µM–9 mM	0.8 µM	CuO/PCL@PPy/ITO	[100]
		Saliva	20–100 µM	20 µM	Co_3_O_4_ needles on Au honeycomb	[65]
		Saliva	0.1 µM–1 mM	0.01 µM	Flexible OECTs- GOx-GO/PANI/Nafion-graphene/Pt	[101]
		Tears, Urine, Saliva	1–10 mM	1 mM	rGO-modified Nb_2_O_5_	[102]
		Tears	0–100 nM	100 nM	GOx-CHIT/Co_3_O_4_ /Au	[103]
		Tears	0–12 mM	9.5 µM	Chitosan-functionalized NG	[104]
		Saliva	0.5 µM–2.5 mM	0.31 µM	IrO_2_@ NiO nanowires	[105]
		Saliva	0.1–8.0 mM	0.02 mM	PEDOT: PSS with Ni/Al LDH	[106]
		Saliva, Sweat	0.1–1000 µM	0.03 µM	AgNPs/MoS_2_	[107]
		Saliva	0.05–2 mmol/L	0.1 mmol/L	Hb-deposited LPG	[108]
		Tears, Urine, Saliva	1–10 mM	1 mM	rGO-modified Nb_2_O_5_	[102]
2	Colorimetry	Tears Saliva	0–47 nM	61 nM 91 nM	Nanoporous palladium (II) bridged coordination polymer	[109]
		Tears, Saliva	0.1–50 mM	1 pM	Pt/Ni@NGT paper-based device	[110]
3	Chemiluminescence	Tears, Saliva	3.0 × 10^−9^–4.0 × 10^−5^ mol L^−1^	6.4 × 10^−10^ mol L^−1^	PG/Co (OH)_2_	[111]
4	Fluorescence resonance energy transfer	Tears	0.03–3 mmol/L	0.03 mmol/L	CdSe/ZnS donor, malachite green dextran acceptor on ZnO nanorods–silicon hydrogel lens	[112]
5	SPR	Urine	8 × 10^−8^–5 × 10^−2^ M	0.8 µM	PMBA@Au/optical fiber with AET/AuNPs	[113]
6	Diffraction spectroscopy	Tears, Blood	0–20 mM	20 mM	PS-MCC	[114]
7	Raman spectroscopy/SPR	Urine	0.01–10^−4^ µM	80 nM	GO-decorated AuNBs@Ag	[115]
8	Photonic band gap	Urine	0 gm/dL to 10 gm/dL	-	Two-dimensional triangular photonic crystal structure	[116]

**Table 4 nanomaterials-13-00867-t004:** List of the most researched nanomaterials for detecting biomarkers, their quantities, and the carbon nanotube media—graphene, nanoparticles, and metal oxides.

S. No.	Nanostructured Materials	Medium	Biomarkers	Concentration	References
1	ZnO Metal nanoparticles Metal Oxides Carbon nanotubes	Blood	Glucose	2–30 mM	[155,156,157]
2	Metal nanoparticles Platinum nanoparticles Carbon nanotubes	Urine	Glucose	2.78–5.5 mM	[158,159,160]
3	Polymer nanostructure Quantum dots Graphene Carbon nanotubes	Saliva	Glucose	0.008–0.21 mM	[160,161,162,163]
4	Polymer nanostructure Carbon nanotubes	Sweat	Glucose	0.277–1.11 mM	[164]
5	Polymer nanostructure Graphene sheet Metal/Metal oxides nanostructures	Tears	Glucose	0.1–0.6	[165,166,167]
6	Polymer nanostructures Metal oxides nanostructures Carbon nanotubes	Breath	Acetone	21–0.5 ppm	[168,169,170,171]

## Data Availability

The data presented in this study are available from the corresponding authors upon reasonable request.

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
