# Peer review of "A Nanotechnology-Based Approach to Biosensor Application in Current Diabetes Management Practices"

_nanomaterials, 2023, doi:10.3390/nano13050867_

Round 1
Reviewer 1 Report
The manuscript entitled, ‘Emerging Role of Biosensors in Current Diabetes Management Practice: A
Nanotechnology Approach’ discussed biosensors in diabetic remediation. The article is nicely discussed but still I am mentioning some points which should be justified before publication;
1. The abstract is too length. Better to shorten with more precise goals.
2. Author should write also about the diabetic types of conventional remediation techniques.
3. Future outlook/perspective should be a specific paragraph which has to be addressed by the authors.
4. Some articles based on sensors have immense significance in this context like: https://doi.org/10.3390/s16040589; https://doi.org/10.1016/j.websem.2023.100774; https://doi.org/10.1016/j.ijbiomac.2019.03.224; https://doi.org/10.1016/j.msec.2018.03.010.
Author Response
Response to decision letter
(Nanomaterials-2228310)
Reviewer 1
The manuscript entitled, ‘Emerging Role of Biosensors in Current Diabetes Management Practice: A
Nanotechnology Approach’ discussed biosensors in diabetic remediation. The article is nicely discussed but still I am mentioning some points which should be justified before publication;
1.The abstract is too length. Better to shorten with more precise goals.
Response: Thank you for your keen observation of our manuscript. As per the suggestion now, the abstract is more oriented toward the title and updated in the revised manuscript.
2. Author should write also about the diabetic types of conventional remediation techniques.
Response: Authors are thankful for the suggestion and modified it as per the requirement in the revised manuscript and highlighted it.
3. Future outlook/perspective should be a specific paragraph which has to be addressed by the authors.
Response: Authors are thankful for the suggestion and we have modified it as per the requirement.
4. Some articles based on sensors have immense significance in this context like: https://doi.org/10.3390/s16040589; https://doi.org/10.1016/j.websem.2023.100774; https://doi.org/10.1016/j.ijbiomac.2019.03.224; https://doi.org/10.1016/j.msec.2018.03.010.
Response: Thank you for your nice suggestion. The information from the given articles has been included, cited, and highlighted in the revised version of the manuscript.

Reviewer 2 Report
The review paper “ Emerging Role of Biosensors in Current Diabetes Management Practice: A Nanotechnology Approach“ by A. Shoaib et al. gives an overview of the different biosensors related to diabetes disease. Starting point of the paper is to summarize the actual situation of diabetes mellitus problems mainly in Saudi Arabia and the reasons for the growing problem for the health situation. After this introduction the different available devices for the diagnosis of diabetes like controlling and monitoring the urine glucose and blood glucose parameters are described. The biosensors used for this are given in an overview manner differentiating in invasive and noninvasive technologies. This is done up to the end of chapter 3 in a very good readable style with extensive references.
Starting with chapter 4 up to chapter 7 the central theme is broken and the manuscript gets somehow strange repeating several statements about bio-nano sensors like what are the needs for good working biosensors. This part has to be rewritten to get a central theme again. There is a lot of information which is more useful as an introduction part stating the advantages of nanotechnology for biosensors and the still open questions for perfect and reliable application of them. Therefore I would suggest to combine these chapters and bring this rewritten part in the front of the others chapters related to special case of biosensors for diabetes monitoring.
The conclusion is again more related to chapter 1 – 3. If you leave out chapter 4 up to 7 this would be another possibility to get a nice paper which meets a paper with a title like “ Emerging Role of Biosensors in Current Diabetes Management Practice”.
Author Response
Response to decision letter
(Nanomaterials-2228310)
Reviewer 2
The review paper “Emerging Role of Biosensors in Current Diabetes Management Practice: A Nanotechnology Approach “by A. Shoaib et al. gives an overview of the different biosensors related to diabetes disease. Starting point of the paper is to summarize the actual situation of diabetes mellitus problems mainly in Saudi Arabia and the reasons for the growing problem for the health situation. After this introduction the different available devices for the diagnosis of diabetes like controlling and monitoring the urine glucose and blood glucose parameters are described. The biosensors used for this are given in an overview manner differentiating in invasive and noninvasive technologies. This is done up to the end of chapter 3 in a very good readable style with extensive references.
Response: The writers are grateful to the reviewer for taking the time and making the effort to go over the manuscript.
Starting with chapter 4 up to chapter 7 the central theme is broken and the manuscript gets somehow strange repeating several statements about bio-nano sensors like what are the needs for good working biosensors. This part has to be rewritten to get a central theme again. There is a lot of information which is more useful as an introduction part stating the advantages of nanotechnology for biosensors and the still open questions for perfect and reliable application of them. Therefore, I would suggest to combine these chapters and bring this rewritten part in the front of the others chapters related to special case of biosensors for diabetes monitoring.
Response: The authors are thankful to the reviewer for taking the time and making the effort to examine our article. We have tried our best to modify the article as per suggestion and updated it in the revised manuscript.
The conclusion is again more related to chapter 1 – 3. If you leave out chapter 4 up to 7 this would be another possibility to get a nice paper which meets a paper with a title like “Emerging Role of Biosensors in Current Diabetes Management Practice”.
Response: We are really thankful for the keen observation of the article. Your suggestions and recommendations are excellent, and we considered this in the revised manuscript.

Reviewer 3 Report
The manuscript addresses an interesting topic. Nevertheless, there are many things that need improvement:
-The title of the manuscript does not match the content.
-The contents are not presented and discussed in a scientifically precise way.
- Some references give information that are outdated. For example, Ref. 1 should be updated.
- Some of the illustrations are so blurry and the fonts are so small that it is impossible to read them.
- Some of the images need to be changed completely. For example, Figure 1 does not represent a biosensor. The signal is an ECG and that is confusing. Figure 4 is not informative.
- The authors should check the formatting of references. E.g. Ref. 2 page reference has wrong format.
- Table 1 has no references. Table 4. part 4: 2 same references
Author Response
Response to decision letter
(Nanomaterials-2228310)
Reviewer 3
The manuscript addresses an interesting topic. Nevertheless, there are many things that need improvement:
Response: We are grateful to you for taking the time and making the effort to examine our article. Your suggestions and recommendations are excellent, and we considered them in the revised version of the manuscript.
-The title of the manuscript does not match the content.
Response: Thank you for your suggestion, we have revised the title accordingly, “A nanotechnology-based approach to biosensor application in current diabetes management practices”.
-The contents are not presented and discussed in a scientifically precise way.
Response: Thank you for your valuable suggestion. As per the suggestions we have modified the manuscript in a precise way.
- Some references give information that are outdated. For example, Ref. 1 should be updated.
Response: Thank you for pointing it out, we have carefully crosschecked all the references and updated accordingly. Reference 1 and 2 have been updated accordingly and highlighted in the revised manuscript.
- Some of the illustrations are so blurry and the fonts are so small that it is impossible to read them.
Response: Thank you for pointing it out, we have carefully revised the quality of images and made similar readable fonts for all captions.
- Some of the images need to be changed completely. For example, Figure 1 does not represent a biosensor. The signal is an ECG and that is confusing. Figure 4 is not informative.
Response: Figure 1 and 4 has been updated and included more information in the revised manuscript.
- The authors should check the formatting of references. E.g. Ref. 2 page reference has wrong format.
Response: Thank you for pointing it out. We have carefully crosschecked all the references and made them all as per journal requirements.
- Table 1 has no references. Table 4. part 4: 2 same references
Response: Thank you, we have carefully checked and updated the references in all the tables included in the revised manuscript.

Reviewer 4 Report
Review report for nanomaterials-2228310
Manuscript ID: nanomaterials-2228310
Manuscript Title: Emerging Role of Biosensors in Current Diabetes Management Practice: A
Nanotechnology Approach.
This manuscript explains about the biosensors and their substantial medical applications. Biosensors in healthcare and diabetes management utilizing nanotechnology. The study also examines numerous diagnostic biosensors for diabetic disorders and unique elements of clinical/allied biosensors, and biosensor clinical practice obstacles. In this approach, this article highlights major advances in nanotechnology-based biosensors for medical applications. It is interesting and is excellent to the readers of the journal. It can be reconsidered after minor revision.
1. The introduction part should have clear motto about this study. This is lack of it.
2. In Table 1, it needs to include the year and events of 2022 and 2023 if it happened.
3. In table 2, you can make two tables and name also different.
4. Need to check English editing.

Author Response
Response to decision letter
(Nanomaterials-2228310)
Reviewer 4
1. The introduction part should have clear motto about this study. This is lack of it.
Response: Thank you. We have updated the introduction section as per the suggestion in the revised manuscript.
2. In Table 1, it needs to include the year and events of 2022 and 2023 if it happened.
Response: Thank you for your suggestion. We have updated Table 1 and included the latest information from the years 2022 and 2023 in the revised manuscript and highlighted it on page no. 12.
3. In table 2, you can make two tables and name also different.
Response: The authors sincerely thank the reviewer for making their review unique and more oriented toward the topic. As per the suggestions, the table is split into two parts, renamed and attached in the revised version of the manuscript.
4. Need to check English editing.
Response: Thank you for your suggestion, we have carefully checked the English (language/grammar/typographical) from the expert.

Round 2
Reviewer 1 Report
This can be published in its present form.
Author Response
Response to decision letter
(Nanomaterials-2228310)
Reviewer 1
- This can be published in its present form.
Response: Thank you and we appreciate your recommendation to consider our manuscript in the present form for publication.

Reviewer 2 Report
The authors changed the manuscript according the advices of the reviewers.
One minor thing: Please check the numbering of the chapters. After chapter 5 chapter 7 is following.
No further comments. Now the manuscript is in a publishable form.
Author Response
Response to decision letter
(Nanomaterials-2228310)
Reviewer 2
The authors changed the manuscript according the advices of the reviewers.
Response: Once again, please accept our sincere thanks for taking the time to review our manuscript and to provide us with valuable suggestions that have made a significant difference in the quality of the article.
One minor thing: Please check the numbering of the chapters. After chapter 5 chapter 7 is following.
Response: Thank you for pointing it out, we have carefully checked and corrected the mentioned chapter no. in the revised version of manuscript.
No further comments. Now the manuscript is in a publishable form.
Response: Thank you for your consideration.

Reviewer 3 Report
Dear authors
The topic of your manuscript has a high impact.
In some aspects, the manuscript needs improvement. It is important that the authors revise the manuscript very carefully:
- It seems that the authors sometimes combine figures from other references without addressing them. For example, part of Fig. 1 and Fig. 3 are from another reference: (Biosensors 2022, 12(11), 927; https://doi.org/10.3390/bios12110927).
- Figure 4 is blurry, the texts in the figure are too small and the figure is not informative.
- Formatting of references is inconsistent, e.g., ref. 2, 4, ... .
Author Response
Response to decision letter
(Nanomaterials-2228310)
Reviewer 3
The topic of your manuscript has a high impact.
In some aspects, the manuscript needs improvement. It is important that the authors revise the manuscript very carefully:
Response: We are grateful to you for taking the time and making the effort to examine our article again. Your suggestions and recommendations are excellent, and we considered them in the revised version of the manuscript.
- It seems that the authors sometimes combine figures from other references without addressing them. For example, part of Fig. 1 and Fig. 3 are from another reference: (Biosensors 2022, 12(11), 927; https://doi.org/10.3390/bios12110927).
Response: Thank you for your keen observation, we have carefully checked the mentioned figures, their references and revised as per the suggestion. The given reference has been cited and highlighted the reference no. in text of the revised manuscript.
- Figure 4 is blurry, the texts in the figure are too small and the figure is not informative.
Response: Thank you for pointing it out. we have carefully revised the quality of figure 4 and made similar readable fonts for all captions.
- Formatting of references is inconsistent, e.g., ref. 2, 4, ... .
Response: Thank you for pointing it out, we have carefully crosschecked all the references, updated according to the format by following consistency.

Round 3
Reviewer 3 Report
Dear authors,
- from the ethic aspect, it is important, if you use figures from other papers or sources in your own manuscript, to mention them appropriately in figure caption or draw the pictures by yourself.
Please consider the above point when revising your manuscript.
- The formatting of references is still inconsistent. Sometimes you write only the first page number of the peper and sometimes the first and the last page number of the paper.
Please take into account the formatting instructions for the references and adjust them accordingly.
Author Response
Response to decision letter
(Nanomaterials-2228310)
Reviewer 3
- from the ethic aspect, it is important, if you use figures from other papers or sources in your own manuscript, to mention them appropriately in figure caption or draw the pictures by yourself.
Please consider the above point when revising your manuscript.
Response: Thank you for your keen observation and for sharing ethical perspective. Regarding the source of the figures from 1 to 4. We accept that all the figures in the manuscript are prepared by us using BioRender software. We consent that all these figures are not taken from any source (paper/internet) as it is the format. Kindly consider the images. Still, at this stage, if you need more clarification, we will be happy to provide you with the same.
- The formatting of references is still inconsistent. Sometimes you write only the first page number of the paper and sometimes the first and the last page number of the paper.
Please take into account the formatting instructions for the references and adjust them accordingly.
Response: Thank you for pointing it out. Please accept our apologies for not following the formatting instructions carefully. The references have now been carefully crosschecked and formatted according to journal guidelines. Please see the revised version of the manuscript.
Once again, please accept our sincere thanks for taking the time to review our manuscript and to provide us with valuable suggestions that have made a significant difference in the quality of the article.
